# Enhancing the Resilience of the Management of Water Resources in the Agricultural Supply Chain

**Wenping Xu [1,2,*], Zhi Zhong [1,2], David Proverbs [3], Shu Xiong [1] and Yuan Zhang [1]**

[1] Evergrande School of Management, Wuhan University of Science and Technology, Wuhan 430065, China; zhongzhi1021@163.com (Z.Z.); xs434991329@163.com (S.X.); zhangyuanzy123@163.com (Y.Z.)
[2] Hubei Industrial Policy and Management Center, Wuhan 430065, China
[3] Faculty of Computing, Engineering and the Built Environment, Birmingham City University, Birmingham B4 7BD, UK; David.Proverbs@bcu.ac.uk
[*] Correspondence: xuwenping@wust.edu.cn; Tel.: +86-177-6252-5770

**Abstract:** Due to population growth and human activities, water shortages have become an increasingly serious concern in recent years. The agri-food industry is the largest water-consuming industry among all industries, and various efforts have been made to reduce the loss of water in the agricultural supply chain. Improving the resilience of water resource management is a key measure to reduce the risks in the agricultural supply chain. This study aims to identify the factors affecting the resilience of water management in the agricultural supply chain and to help manage the risks related to water resources use. A total of 14 factors are selected from five dimensions of society, economy, environment, institution, and crop characteristics, and an index institution is constructed. This was used to determine the level and importance of each factor. Data were collected through a questionnaire survey involving 28 experts from the agricultural industry in Northwest China, as well as a thorough literature analysis and interactions with experts. Using a combination of an interpretative structural model (ISM) and an analytical network process (ANP), a hierarchical structure model was developed, composed of direct factors, indirect factors, and basic factors. The results show that the selection of agricultural products, the establishment of a water audit control institution, the recycling of wastewater, and the investment in water-saving technologies are the main factors affecting the resilience of water resources management in the agricultural supply chain. These findings provide useful guidance for practitioners involved in the management of water resources in agricultural supply chains. These results are expected to contribute to the sustainable management and strategic deployment of water by agricultural supply chain stakeholders.

**Keywords:** water management resilience; agricultural supply chain; ISM-ANP model

## 1. Introduction

With the growth of population and human activities, the shortage of water resources has become a more serious concern in recent years. Many countries and regions are facing the serious problem of water shortages, which not only poses a great threat to commercial activities but also poses a great threat to human beings [1]. The scarcity of water resources seriously restricts the development of human society. As the largest freshwater user in industry, agriculture accounts for more than 70% of global freshwater consumption among all water-consuming sectors. In addition, inefficient water-use patterns also exacerbate the conflict between water demand and water supply in the agricultural water management system [2]. Water is the main resource of the agricultural food industry, and pressures to reduce consumption in line with conserving environmental and natural resources are considered to be the dominant driving force of agricultural water management in the food supply chain [3]. How to ensure the realization of agricultural water savings on the basis of food security has become a major global concern in the 21st century

[4] within the water-intensive industries [5]. In the context of water scarcity, water resource management has attracted the interest of various disciplines, and the role of the private sector in integrated water resource management in the agri-food supply chain is growing [6]. In addition, the concept of water management has further aroused private sector interest in integrated water management, in which the role of corporations as water managers transforms the global water governance landscape by participating in water management and mitigating the negative environmental and social impacts of their supply chains. Water management has been identified as a new framework for enterprises to participate in water resource management [7]. With the improvement of consumers' awareness of environmental protection, consumers' demand for water-saving products increases, thus making enterprises more responsible for their water resource management [8]. At the same time, a series of tools, assurance, and certification programs related to enterprise water resource management has emerged. This reflects the increased awareness of the operational, environmental, and reputational risks in the corporate sector, and this trend is accompanied by increasing pressure from multiple stakeholders to demonstrate transparency in water management [9].

The current production systems in the agri-food industry are highly dependent on water [10]. For the company, there is a drive to improve profitability and competitiveness from the management of water, as efficient water management helps to reduce production costs [11]. Water management practices can also improve the participation of the various stakeholders in the supply chain, encourage not only the companies themselves but also suppliers, customers, Non-Governmental Organization (NGOs), and the public sector to participate in water management practices [12]. Policy pressures are seen as another major driver in the agricultural food supply chain. Policymakers tend to formulate new policies, urging food companies to pay more attention to the management of water in agriculture in order to improve its sustainable supply chain performance [13]. The commission stressed that agriculture needs a more flexible approach to better cope with the current and future economic, social, and environmental challenges. Hence, supporting the resilience of agricultural systems has become an important objective of the post-2020 Common Agricultural Policy recommendations [14].

In terms of agricultural water resources management, Jellasonn [15] systematically analyzed 1086 articles by Scopus and Google scholars and found that long-term arid desertification and water dryness are common features of long-term challenges for smallholders to achieve resilience and agricultural sustainability in arid regions. Through a statistical analysis of the water resources in Tunisia, an arid country in the Mediterranean Sea, Ahmed [16] found that the use of unconventional water resources (saltwater and treated wastewater) has become very urgent. The selection of water-saving and drought-resistant, and saline-alkali resistant varieties through the drip irrigation system is very important for agricultural development. Yu [17] conducted a meta-analysis of global drylands (81 studies and 836 paired observations) to evaluate the response of various crops to drought and found that improving crop water-use efficiency can ensure the sustainable development of food production in drylands. Wang [18] used the results of a Tobit model to show that agricultural investment and production, economic growth, industrial structure adjustment, and agricultural plant structure adjustment have important effects on agricultural water-use efficiency.

Water-use efficiency (WUE) directly affects the water consumption of agricultural production and is of great importance to local and regional water savings. Agricultural water-use efficiency is also a key indicator reflecting the effective allocation of water and the improvement of water productivity in different agricultural sectors. These research results can provide important references for agricultural water management in the middle reaches of the Huaihe River Basin and other similar areas in Northwest China. Hadizadehf conducted a survey on paddy farmers, using exploratory factor analysis, revealing the influence of rice farmers on the agricultural water management considering five factors: (i) the usability of the irrigation infrastructure, (ii) planting patterns, (iii) the support of

local institutes, (iv) irrigation experience, and (v) traditional beliefs. These factors combined accounted for 60.1% of the total balance of water management in agriculture. These findings provide a better understanding of the drivers of integrated agricultural water management by paddy farmers and help policymakers focus on strategies to improve irrigation water productivity and support more sustainable water use in rice production in the study area and in similar drought-crop regions around the world [19].

Wu modeled the Borley Ecoecosystem Productivity Simulator with remote sensing data and observation data of ground stations as input and believed that drought index based on remote sensing data could promote dynamic agricultural drought assessment, and the obtained drought index could provide dynamic information for real-time monitoring. These results can provide important references for agricultural water resources management in arid areas [20]. Guiqin, based on gray relation analysis (GRA), developed a method to estimate the agricultural water vulnerability and identified the main factors that influence the development of agricultural drought susceptibility [21]. Ridouttb believes that in the life cycle of a product, the primary production stage often has the greatest impact on water resources, but it should also consider the interaction between different stages of the supply chain and how the company's role in water management affects its supply chain, as global supply chains are becoming more and more complex. The impact on water resources is often far away from the final consumer of products [22]. A variety of water conditions can lead to an imbalance of space and time distribution of water resources and have a profound influence on the risk of water shortage. In order to meet our demand for water resources, changes to the water supply in space and time are necessary to determine the critical path of resilience and the critical point at which the natural freshwater system is reformed [23,24].

Water governance continues to be a challenge for human society, with the increasing scale and frequency of adverse events caused by climate and anthropogenic change and the occurrence of crises in water resources systems [25,26]. Through planning the use of water resources, it is found that there is a potential synergistic effect of water-use planning on water resilience [27]. The interaction between water managers, users, and water components affects the implementation of water planning [28]. On the supply side of water resources, economic capacity and rapid access to funds are the main economic factors affecting the resilience of water supply systems [29]. Drip irrigation, which has been widely used in arid areas in recent years, can make an important contribution to more sustainable water use in drought-prone areas, but the autonomy of localized irrigation systems needs more attention from local governments [30,31]. Liu used the support vector machine model based on the improved gray wolf optimization algorithm (IGWO-SVM) to evaluate the resilience of agricultural irrigation in a severe cold region and put forward targeted suggestions for local water resources management [32].

In terms of water resilience, Hashimotot is the first to assess the sustainability of water systems using traditional reliability, resilience, and vulnerability (RRV) criteria. These performance criteria refer to how likely the system is to fail (reliability), how severe the consequences of failure (vulnerability) are, and how quickly it can recover from failure (resilience) [33]. Resilience is defined as the ability of a water management system to "bounce back", that is, absorb and then recover from water scarcity events and return to normal system functioning [34]. The concept of resilience has become increasingly prominent in water policy and research over the past decade [35]. Resilience criteria denote the ability of water resource systems to absorb the impact of an event and return to an acceptable operational condition after a disturbance. These performance criteria refer to how likely a system is to fail (reliability), how severe its consequences are (vulnerability), and how quickly it can recover from failure (resilience) [36]. Imanim developed a new application using artificial neural networks (ANN) to predict water quality resilience and simplify resilience assessment [37].

Kharrazia examines system-level configurations and trade-offs related to water resource resilience management using a holistic approach called ecological network analysis (ENA) [38]. Xue B. [39] investigated different crops functional types of drought field level and watershed hydrology resilience and found the hydrological resilience of crops is related to drought intensity and water-use efficiency. These research results can provide important references for crop water efficiency and the choice of crops in arid regions [38,39]. Royr established a framework for agricultural resilience that includes three capacities (absorptive, adaptive, and transformative) and five dimensions (social, economic, ecological, physical, and institutional). Using a combination of top-down and bottom-up approaches, 15 indicators were developed to assess the resilience of coastal agricultural systems that were used to develop a strategy for the management of coastal agricultural systems in Bangladesh [40]. Lim developed a risk-based interval optimization modeling method for agricultural water allocation in view of the complexity of uncertainty and risks in agricultural water management systems. The method includes a conditional value at risk (CVaR) model, a two-stage stochastic programming (ITSP) model with inexact probability (IPS), and a stochastic boundary interval (RBI) in general framework. This method can balance the expected benefits, penalties, and risks of agricultural water allocation at the same time, solve the uncertainty of agricultural water supply and demand in the form of probability distribution and random boundary interval [41].

Behboudian, M. [36] used a new method for quantifying the total resilience of water management scenarios. The effects of climate change on water supply and demand were investigated using calibrated soil and water assessment tools and water distribution models. Water resource system resilience is measured from five aspects. The first aspect defines resilience as the strength of the system to resist crossing performance thresholds (reliability). In the second aspect, if the system exceeds the performance threshold, the recovery rate of the system after the disturbance is assessed. Violations of the performance threshold have been factored into a third dimension (vulnerabilities), which takes into account the severity of the failure. The fourth dimension is resilience to extreme events with unknown probability, which includes four sub-criteria, namely speed, robustness, resourcefulness, and redundancy (4R). The fifth criterion takes into account the ecological status of the system (ecological index). To compare water resource management options (alternatives), a method based on analytical evidence reasoning (ER) was used [42]. NAMW proposed a practical method to assess water supply vulnerability and sustainability by using climate change based on time-dependent analysis of water supply and demand and applied the vulnerability assessment model to evaluate and predict the potential impact of agricultural water demand and supply on reservoir operation, so as to improve local water management under climate variability and change [43]. Dardon Villem [44] believes that susceptibility, resilience, robustness, and adaptability are the four key concepts of system dynamics in the event of a disturbance. However, making them operable for agricultural systems using quantitative dynamic methods remains a challenge [43,44].

According to previous research results, work on water resources management in the agricultural supply chain is mainly concerned with the agricultural water-use efficiency and mainly focuses on the management of water reduction in specific stages of the agricultural supply chain (usually primary production), for example, using drip irrigation technology and wastewater recycling technology. The driving factors for water management in the agricultural supply chain are mainly the background of water shortage, policy pressures, and private sector participation. Moreover, the imbalance of the spatial and temporal distribution of water resources encountered by various water sources has a profound impact on the risk of water shortage. The artificial change of water supply in space and time is the key approach to resilience, and there is a potential synergistic effect on the planning and use of water resources and water resilience. The main research on the resilience of water resources is to evaluate the resilience of water resources by establishing various resilience frameworks, but the quantitative assessment of the resilience of water

resources management has not been introduced into the agricultural supply chain. Furthermore, there is also a lack of a comprehensive framework for the resilience of water resources management of the entire agricultural supply chain.

To fill this gap, this paper focuses on improving water management resilience at multiple stages of the agri-food supply chain. A conceptual framework for integrated water management in the agricultural supply chain is proposed by summarizing the main findings of current research, combining agricultural water management with resilience, and taking agriculture in Northwest China as an example. Through the combination of ISM and ANP, a multi-level hierarchical structure model composed of direct factors, indirect factors, and basic factors is obtained. The study considers the agricultural water management supply chain, moving from a single phase to multiple stages and moving from a focus on agricultural water-use efficiency to consider more widely the resilience of agricultural water resources.

## 2. Methods

The interpretative structural model (ISM) allowed the internal structure of a system to be revealed by processing known but messy system element relationships and was put forward in 1973 by Warfield [45]. Its basic principle is to decompose the constituent elements of a complex system into several sub-elements. Drawing on a combination of theoretical knowledge, practical experience, and statistical analysis, the system elements are made into a multi-level hierarchical structure diagram. Analytical Network Process (ANP) is evolved from the Analytic Hierarchy Process (AHP), which is applied to assign weights of the selected dimensions and indicators. AHP process provides weights to indicators by pair-wising dimensions and indicators without considering interdependent relationships among dimensions. To deal with the uncertainty of interdependency of indicators and complex network relationship of dimensions, ANP can be used to determine the mixed weight [46, 47].

In this study, firstly, an evaluation index of agricultural supply chain water resource management resilience was developed, including the direct factors, indirect factors, and deep-seated fundamental factors by using ISM. Then, based on the multi-level hierarchical structure obtained by ISM, the ANP structure chart was established. To establish the importance of the evaluation indexes that affect agricultural supply chain water resource management resilience, the relative weight of each evaluation index was obtained by using Super Decisions software. The specific principles are shown in Figure 1.

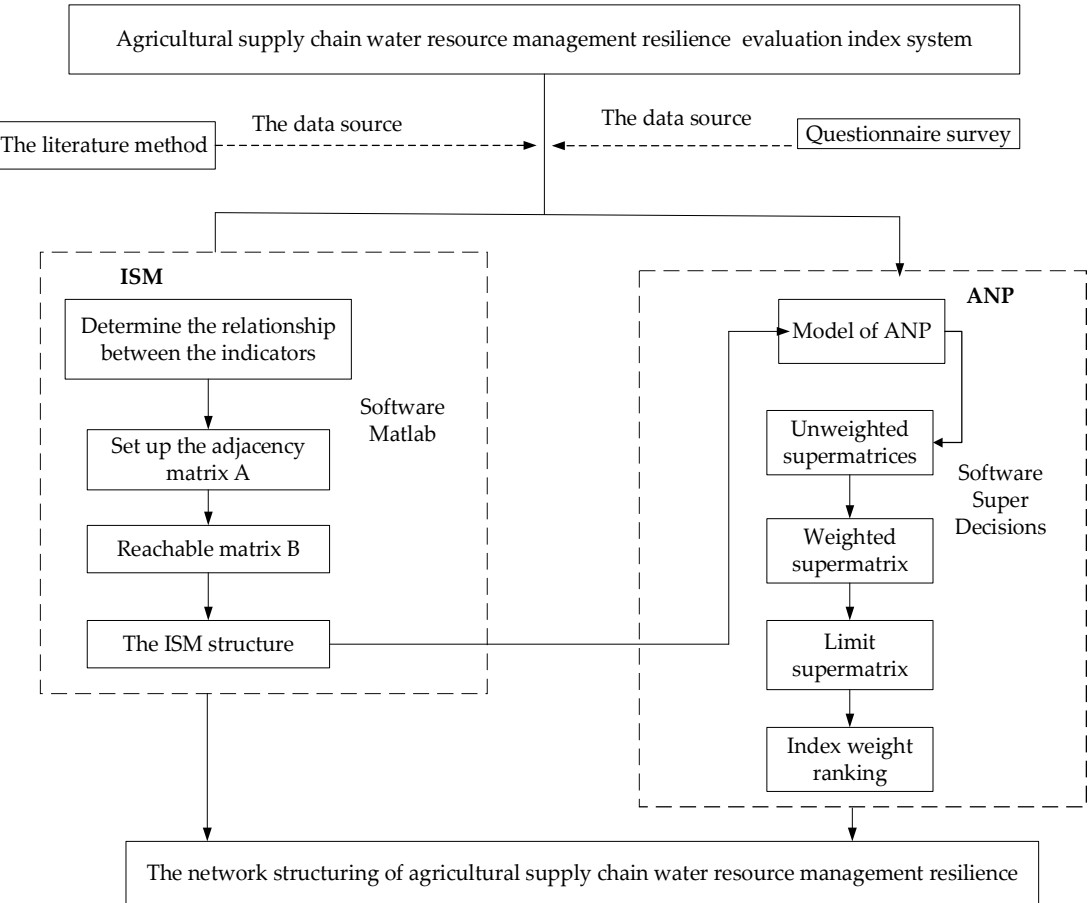

**Figure 1.** Factor analysis of agricultural supply chain water resource management resilience based on ISM-ANP.

### 2.1. ISM Method

Step1: Establish an adjacency matrix.

The factors found to influence the management of water resources in the agricultural supply chain resilience are denoted as $Y_1$, $Y_2$,..., $Yn$, $n$ is the quantity of the resilience influencing factors, and $Y$ is the set of resilience influencing factors. The directed graph $G$ described as a mathematical formula:

$$G = \{(Y, R)|Y = n, R = m\} \tag{1}$$

where $Y = \{Y_1, Y_2 ..., Y_n\}$ and $R = \{(Y_i, Y_j)|Y_i, Y_j \epsilon Y\}$ The directed graph model describes the interrelationship between the elements of the influencing factors. $A$ directed graph model was created to construct the adjacency matrix and the reachable matrix.

The factors were compared using the Delphi method to attribute scores to represent the degree of influence of each factor and to establish the adjacency matrix. The relationship between two factors in a directed graph $G$ can be represented by an $n \times n$ adjacency matrix.

$$A = (a_{ij}) \times a_{ij} = \begin{cases} 1, & (Y_i, Y_j)\epsilon R \\ 0, & \text{other} \end{cases} \tag{2}$$

Step 2: Calculate the reachability matrix $B$.

The reachable matrix $B$ can be obtained by processing adjacency matrix $A$ with Boolean operation rules.

$$B = (A + I)^{n+1} = (A + I)^n \neq \cdots \neq (A + I)^2 \neq (A + I) \tag{3}$$

The reachability matrix reflects the structural relationship between the influencing factors after continuous influence.

Step 3: Decomposition reachability matrix $B$.

By decomposing the reachable matrix to construct the structural analysis model, a hierarchical structure diagram was established.

### 2.2. ANP Method

The structure of ANP is recursive and involves a combination of hierarchies in which levels and inner loops dominate each other. The ANP system consists of two parts: the control layer and the network layer. The control layer is composed of objectives and criteria. Each criterion hypothesis is independent of the other and is only influenced or dominated by the target. Therefore, if there are criteria when determining the weight of each criterion, it can be solved by the AHP method. In an ANP system, objectives are required, but criteria may not be necessary. The clusters in the network layer are controlled by the control layer, and the clusters in the network layer also influence each other. Each element in the network layer inner cluster also influences each other. The analytical network process (ANP) structure is shown in Figure 2.

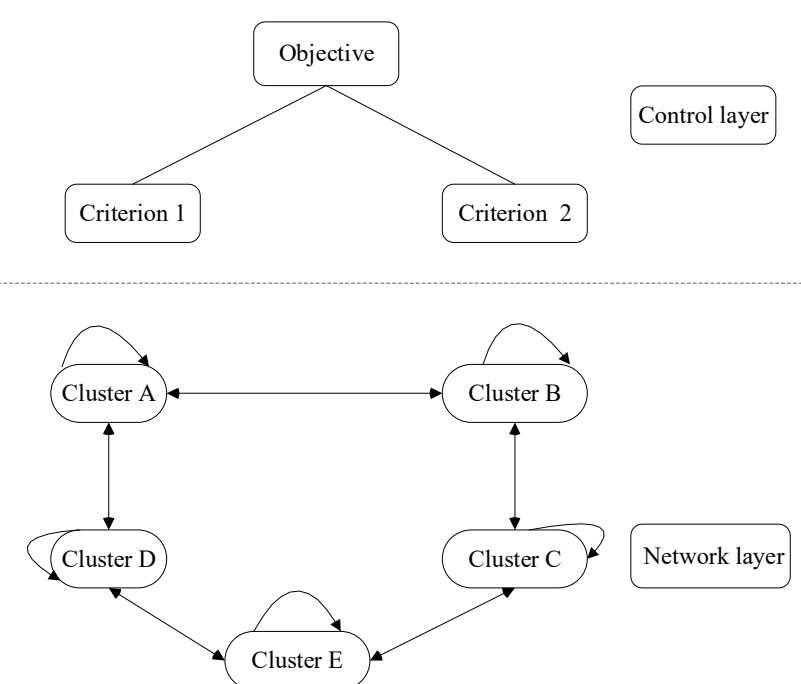

**Figure 2.** Analytical Network Process (ANP) structure.

The ANP could be described as follows:

1. According to the comprehensive set of criteria specified by the experts, the first step was to identify the relationship between the criteria, sub-criteria, and alternatives that are displayed in a graphical network structure. The relationship identified at this step can be both between and within clusters [48,49].
2. Pairwise coupling the comparison matrices of levels and indicators: Decision indicators at each dimension are compared pairwise with respect to their importance toward the same dimensions, and the dimensions themselves are also compared pairwise regarding their contribution to resilience. The relative importance values are determined by Saaty's scale as in AHP (Table 1), and then, the priority vector can be calculated.
3. The special vector was calculated in Equation (4). In order to check the inconsistency rate, $\lambda$ max was calculated using Equation (5), *CI* the consistency indicator Equation (6), *RI* the random indicator, and *CR*, the consistency rate calculated according to Equation (7). The random indicator is extracted from the standard random indicator table. In addition, the consistency rate was set at an amount less than 0.1 [48].

$$W' = AW = \begin{bmatrix} w'_1 \\ w'_2 \\ \vdots \\ w'_n \end{bmatrix} \tag{4}$$

$$\lambda\text{max} = \frac{1}{n}\left(\frac{w'_1}{w_1} + \frac{w'_2}{w_2} + \cdots \frac{w'_n}{w_n}\right) \tag{5}$$

$$CI = \frac{\lambda\text{max} - n}{n - 1} \tag{6}$$

$$CR = \frac{CI}{RI} \tag{7}$$

4. Supermatrix formation and selection of the best alternatives.

The unweighted supermatrix was constructed by replacing the internal priority vectors (relative weights), the elements, and clusters. Then, there was an obligation to standardize the unweighted supermatrix to sum up each column and construct a weighted supermatrix. To calculate the limited super matrix, according to Equation (8), the weighted supermatrix has to be raised enough to a larger power in order to produce convergence; that is, all elements of each row are needed to be identical [50].

$$\lim_{z \to \infty} W^2 w \tag{8}$$

**Table 1.** Saaty's scale.

| Scale of Importance | Linguistic Term | Explanation |
|:---:|:---:|:---:|
| 1 | Equal importance | Two indicators contribute equally to the objective |
| 3 | Moderate importance | Judgment slightly favor one indicator over another |
| 5 | Strong importance | Judgment strongly favor one indicator over another |
| 7 | Very strong | An indicator is favored very strongly over another |
| 9 | Extreme strong | An indicator is favored extremely strongly over another |

## 3. Factors Influencing Water Resources Resilience in Agricultural Supply Chain

### 3.1. Study Area

Due to the mismatch between the allocation of water resources and the space for food and energy production, 80% of the water resources are in the south of China, and 65%of the arable land is in the north. This presents severe challenges for China in balancing the development of the economy and the ecosystem. Over the past few decades, the northwest region has played a prominent role in safeguarding the country's food and energy security. However, the lack of water resources in the region poses a huge threat to the sustainable development of water resources in the northwest while enhancing the resilience of water resources in the west. Hence, Therefore, this paper chooses Shaanxi and Ningxia in Northwest China as the target areas, as shown in Figure 3.

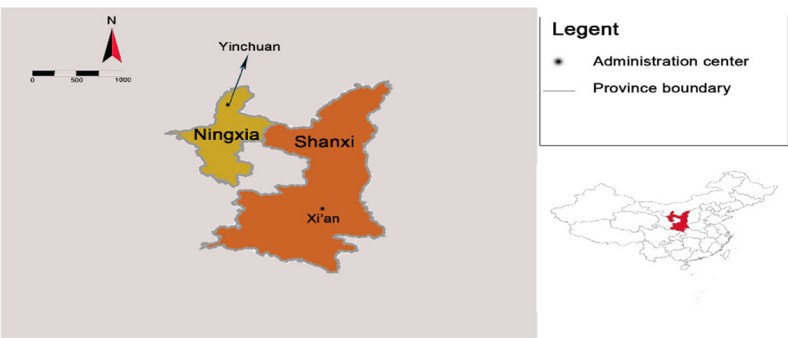

**Figure 3.** Map of the study area.

*3.2. Data Collection and Factor Selection*

Data collection commenced with a survey of experts involving a total of 30 experts, including agricultural supply chain practitioners as well as academic experts in the relevant field. The participants were selected using the purposive sampling method [51]. Purposive sampling is also known as subjective or judgmental sampling method as it relies on the judgment of the researchers when choosing the members of the population for the study. To address this, the experts were chosen based on key characteristics of interest. Table 2 represents a brief description of the profiles of these experts. The questionnaire was sent to 30 experts via e-mail, resulting in 28 valid responses, representing a 93% response rate. The introduction of 28 experts is shown in Table 2. The questionnaire survey is shown in Appendix A**.**

**Table 2.** Brief description of profiles of the experts.

| Characteristics | Position | Number | Specialty | Years |
|---|---|---|---|---|
| Professor | University professor | 2 | Environment, water resources management | More than 12 years |
| | University associate professor | 2 | Risk management |  |
| | | | Resilience management | |
| Chief Operating Officer | Local chief operating manager of listed agricultural products company | 2 | Agricultural supply chain resilience management | 8 to 12 years |
| | Operation manager of local crop seed enterprise | 2 | | |
| | Agricultural product operation manager of private enterprise | 2 | | |
| | Operation manager of state-owned agricultural products enterprise | 2 | | |
| Government Agricultural Water Management Agency | Office director | 3 | Resilience management | 8 to 12 years |
| | Deputy director of office | 2 | Resilience management/Risk management | |
| | Clerk | 1 | Risk management | Up to 8 years |

| Production Management | Wheat processing manager in a state-owned enterprise | 1 | Supply chain management | |
| | Agricultural product manager of listed company | 2 | Production management | |
| Supply Chain Management | Operation manager of private agricultural enterprises (supply chain direction) | 1 | Supply chain management (water resources) | 8 to 12 years |
| | Operation manager of state-owned agricultural products enterprise (supply chain direction) | 6 | | |

Through a combination of the results of a questionnaire survey, a practical investigation of an agricultural produce company based in Shaanxi, and a review of related literature, 14 factors found to influence agricultural supply chain water management resilience were selected on the basis of the common framework [6,9]. These were drawn from the five dimensions of agricultural supply chain water management, namely: society, economic, environment, institution, crop factors, as shown in Table 3.

**Table 3.** Multiple influencing factors of agricultural supply chain water management resistance.

| | **Criterion Layer** | **Secondary Indicator Layer** | **Description** |
|---|---|---|---|
| Index system of influencing factors of agricultural supply chain water management resilience (A) | Society (X1) | Water-saving awareness (S1) | Consumers' awareness of water conservation |
| | | | Farmers' awareness of water conservation |
| | | | Water-saving awareness of enterprises |
| | | Reduce waste (S2) | Reduce waste of agricultural products |
| | | | Reduce water waste in agriculture |
| | | Identification of water-use risks (S3) | Drought and flood disaster risk |
| | | | Water cost risk |
| | Economic (X2) | Investment in water-saving technologies (S4) | Invest in drip irrigation technology |
| | | | Investment in rainwater harvesting systems |
| | | Stakeholder involvement (S5) | Government |
| | | | Enterprise |
| | | | Consumers |
| | | | Farmers |
| | | Reasonable water price (S6) | Water consumption is priced in stages |
| | Environment (X3) | Use chemicals with caution (S7) | Use of insecticides |
| | | | Fertilizer use |

| | | |
|---|---|---|
| | Improve water retention in the soil (S8) | Soil improvement |
| | | Enhanced organic content |
| | Wastewater recovery and use (S9) | Industrial wastewater |
| | | Agricultural wastewater |
| | | Life wastewater |
| Institution (X4) | Public sector water management policy (S10) | The water rights trading |
| | | Water balance for agriculture and industry |
| | Establish water audit control system (S11) | Ecological audit |
| | | Economic audit |
| | | Compliance audits |
| | Integrate water management into business strategy (S12) | Improve the environmental reputation of enterprises |
| | | Improve operational efficiency |
| | | Reduce pressure from competitors |
| Crop characteristics (X5) | Improving traditional crops (S13) | The use of hybridization techniques |
| | | The use of transgenic technology |
| | Selection of crops (S14) | Plant crops that require less water |
| | | Drought-resistant crops |

Based on a combination of the findings from Table 3 and related evidence from the literature, an evaluation model of agricultural supply chain resilience to water management was established, as shown in Figure 4. Figure 4 shows the water resource resilience model of the agricultural supply chain under five dimensions. The five criteria layers are social, economic, environment, institution, and crop characteristics and include 14 indicators of water resource resilience as the secondary indicators in the agricultural supply chain.

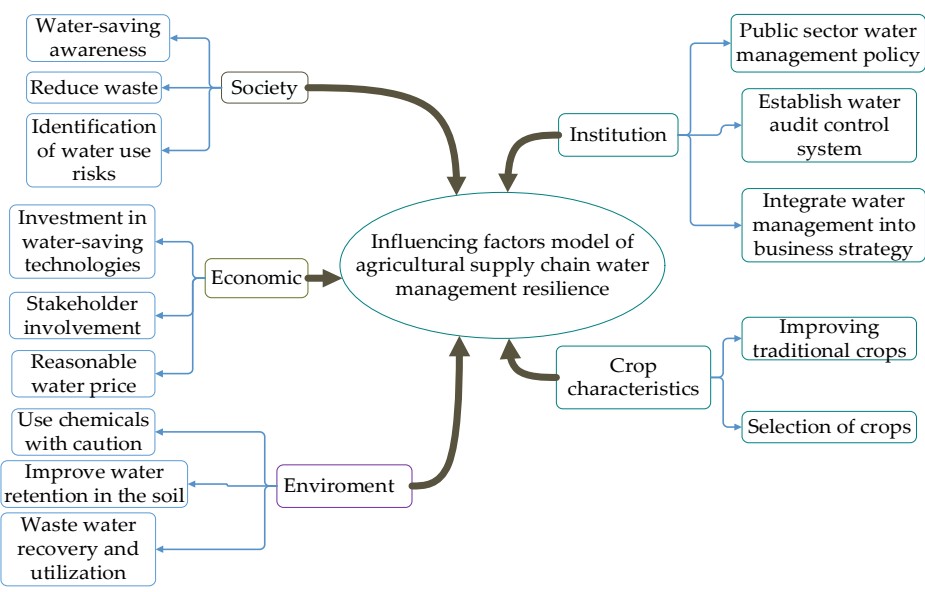

**Figure 4.** A model of factors found to influence agricultural supply chain resilience to water.

### 3.3. Key Factors Analysis

Using the ISM method, the data of the influence relationships were obtained by questionnaire survey with the final data for each item being the highest frequency at which all expert results were selected and leading to the adjacency matrix A as shown in Table 4.

**Table 4.** Adjacency matrix A.

|     | S1 | S2 | S3 | S4 | S5 | S6 | S7 | S8 | S9 | S10 | S11 | S12 | S13 | S14 |
|-----|----|----|----|----|----|----|----|----|----|-----|-----|-----|-----|-----|
| S1  | 0  | 1  | 0  | 1  | 0  | 0  | 1  | 1  | 1  | 0   | 0   | 0   | 0   | 1   |
| S2  | 0  | 0  | 0  | 0  | 0  | 0  | 0  | 0  | 0  | 0   | 0   | 0   | 0   | 0   |
| S3  | 0  | 1  | 0  | 0  | 0  | 0  | 1  | 1  | 1  | 0   | 0   | 0   | 0   | 1   |
| S4  | 0  | 0  | 0  | 0  | 0  | 0  | 0  | 0  | 0  | 0   | 0   | 0   | 0   | 0   |
| S5  | 0  | 0  | 1  | 1  | 0  | 1  | 1  | 0  | 1  | 1   | 0   | 0   | 0   | 0   |
| S6  | 0  | 1  | 0  | 0  | 0  | 0  | 0  | 1  | 1  | 1   | 0   | 0   | 1   | 0   |
| S7  | 0  | 0  | 0  | 0  | 0  | 0  | 0  | 0  | 0  | 0   | 0   | 0   | 0   | 0   |
| S8  | 0  | 0  | 0  | 0  | 0  | 0  | 0  | 0  | 0  | 0   | 0   | 0   | 0   | 0   |
| S9  | 0  | 1  | 0  | 1  | 0  | 0  | 1  | 1  | 0  | 0   | 0   | 0   | 0   | 0   |
| S10 | 0  | 0  | 0  | 0  | 0  | 0  | 0  | 0  | 0  | 0   | 0   | 0   | 0   | 0   |
| S11 | 0  | 1  | 1  | 0  | 0  | 1  | 1  | 0  | 1  | 0   | 0   | 0   | 1   | 1   |
| S12 | 0  | 0  | 1  | 1  | 0  | 0  | 1  | 1  | 1  | 0   | 0   | 0   | 1   | 1   |
| S13 | 0  | 1  | 0  | 0  | 0  | 0  | 1  | 0  | 0  | 1   | 0   | 0   | 0   | 1   |
| S14 | 0  | 0  | 0  | 0  | 0  | 0  | 0  | 0  | 0  | 0   | 0   | 0   | 0   | 0   |

According to the adjacency matrix, the relationship between the factors is obtained, and the reachability matrix B between indexes is obtained by Boolean operation, as shown in Table 5.

**Table 5.** Reachable matrix B.

|     | S1 | S2 | S3 | S4 | S5 | S6 | S7 | S8 | S9 | S10 | S11 | S12 | S13 | S14 |
|-----|----|----|----|----|----|----|----|----|----|-----|-----|-----|-----|-----|
| S1  | 1  | 1  | 0  | 1  | 0  | 0  | 1  | 1  | 1  | 0   | 0   | 0   | 0   | 1   |
| S2  | 0  | 1  | 0  | 0  | 0  | 0  | 0  | 0  | 0  | 0   | 0   | 0   | 0   | 0   |
| S3  | 0  | 1  | 1  | 1  | 0  | 0  | 1  | 1  | 1  | 0   | 0   | 0   | 0   | 1   |
| S4  | 0  | 0  | 0  | 1  | 0  | 0  | 0  | 0  | 0  | 0   | 0   | 0   | 0   | 0   |
| S5  | 0  | 1  | 1  | 1  | 1  | 1  | 1  | 1  | 1  | 1   | 0   | 0   | 1   | 1   |
| S6  | 0  | 1  | 0  | 1  | 0  | 1  | 1  | 1  | 1  | 1   | 0   | 0   | 1   | 1   |
| S7  | 0  | 0  | 0  | 0  | 0  | 0  | 1  | 0  | 0  | 0   | 0   | 0   | 0   | 0   |
| S8  | 0  | 0  | 0  | 0  | 0  | 0  | 0  | 1  | 0  | 0   | 0   | 0   | 0   | 0   |
| S9  | 0  | 1  | 0  | 1  | 0  | 0  | 1  | 1  | 1  | 0   | 0   | 0   | 0   | 0   |
| S10 | 0  | 0  | 0  | 0  | 0  | 0  | 0  | 0  | 0  | 1   | 0   | 0   | 0   | 0   |
| S11 | 0  | 1  | 1  | 1  | 0  | 1  | 1  | 1  | 1  | 1   | 1   | 0   | 1   | 1   |
| S12 | 0  | 1  | 1  | 1  | 0  | 0  | 1  | 1  | 1  | 1   | 0   | 1   | 1   | 1   |
| S13 | 0  | 1  | 0  | 0  | 0  | 0  | 1  | 0  | 0  | 1   | 0   | 0   | 1   | 1   |
| S14 | 0  | 0  | 0  | 0  | 0  | 0  | 0  | 0  | 0  | 0   | 0   | 0   | 0   | 1   |

The interpretative structure model diagram of multi-level hierarchical structure is obtained by dividing the levels. The multi-level hierarchical structure is composed of direct factors in the surface layer, indirect factors in the middle level, and fundamental factors in the deep layer and was obtained by an interpretative structure model (ISM), as shown in Figure 5.

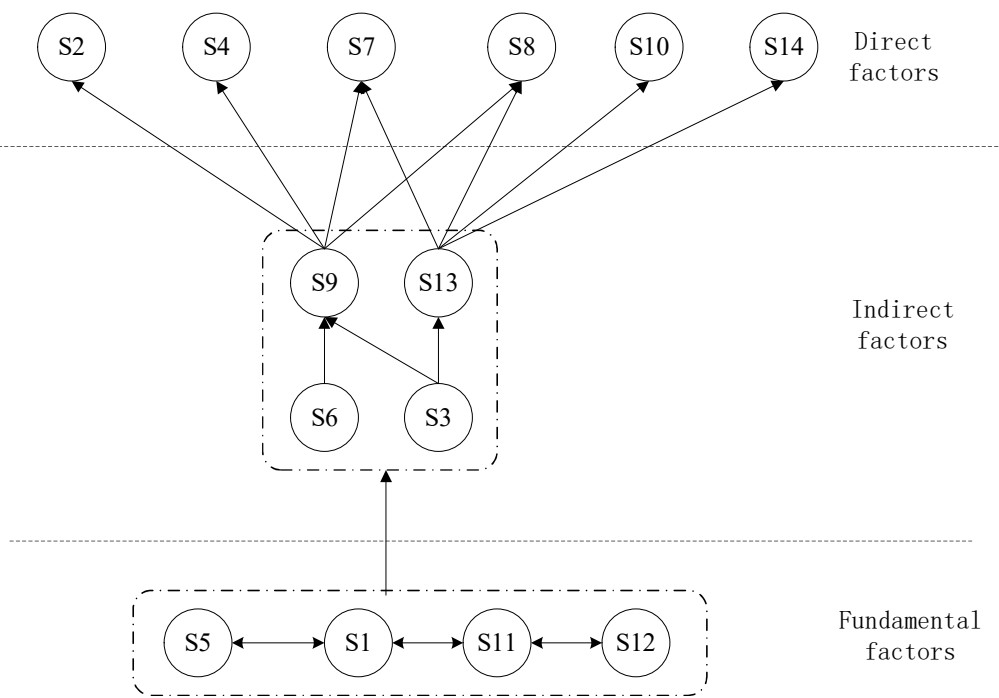

**Figure 5.** Multi-layer hierarchical diagram of influencing factors.

According to Figure 5, the hierarchy of charts clearly shows the elements and interactions of water management resilience in the agricultural supply chain. Directly on the surface of the factors including waste reduction, investment in water-saving technology, careful use of chemicals, improvement of the water retention in the soil, the public sector water management policy, selection of crops. Indirect factors including wastewater recycle use, improved conventional crops in water, reasonable water price formulation, identify risks as factors that are indirectly affected by the water management resilience. At the same time, it also reflects the constraints of the surface influencing factors. The basic factors include the participation of social stakeholders, the awareness of water saving, the establishment of water audit control systems, and the integration of water management into enterprise strategy, which reflects the root and essence of water resource management resilience factors in the agricultural supply chain.

The ANP model is built on the basis of the relationship between ISM indicators, as shown in Figure 6. Before calculating the weightless supermatrix, a judgment matrix should be constructed. After consulting experts, the judgment matrix for the pairwise comparison of each indicator was developed, and these judgments were entered into the Super Decisions software to calculate the weightless. A column is a sorting weight based on the element. If there is no effect, the value is 0. Based on Super Decisions software, the judgment matrix was constructed and calculated to check consistency. Based on the analytical network process, the total weight of each factor index is calculated, the overall normalized weights of the evaluation indicators of the resilience of management of water resources in the agricultural supply chain as shown in Table 6. In the next stage, the supermatrix for compatibility is formed from converging to a long-term and stable set of weights, which presents the results of relative importance, as shown in Tables A1 and A2. The supermatrix for compatibility after convergence is indicated in Table A3.

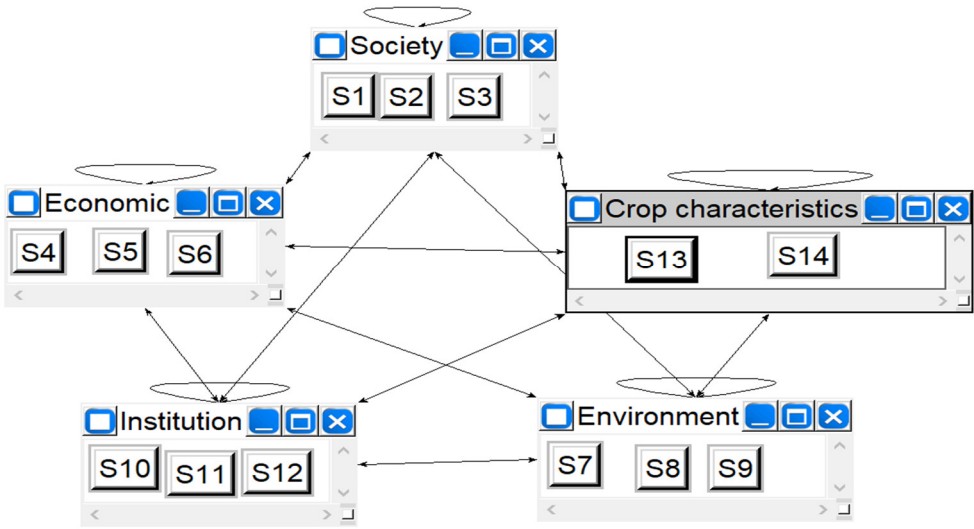

**Figure 6.** Analytical Network Process model.

**Table 6.** The index factor weights of the indicator.

| Criterion Layer | Secondary Indicator Layer | Score | Total Score |
|---|---|---|---|
| | S1 | 0.050137 | |
| X1 | S2 | 0.014621 | 0.14798 |
| | S3 | 0.083222 | |
| | S4 | 0.088506 | |
| X2 | S5 | 0.071831 | 0.199578 |
| | S6 | 0.039241 | |
| | S7 | 0.012268 | |
| X3 | S8 | 0.057894 | 0.172515 |
| | S9 | 0.102353 | |
| | S10 | 0.017648 | |
| X4 | S11 | 0.163096 | 0.250984 |
| | S12 | 0.07024 | |
| X5 | S13 | 0.072417 | 0.228943 |
| | S14 | 0.156526 | |

According to the index factor weights obtained in Table 6, the total weight comparison of each index in each dimension of water resource management resilience in the agricultural supply chain is shown in Figure 7, and the weight ratio of each index in each dimension of water resource management resilience in the agricultural supply chain is shown in Figure 8.

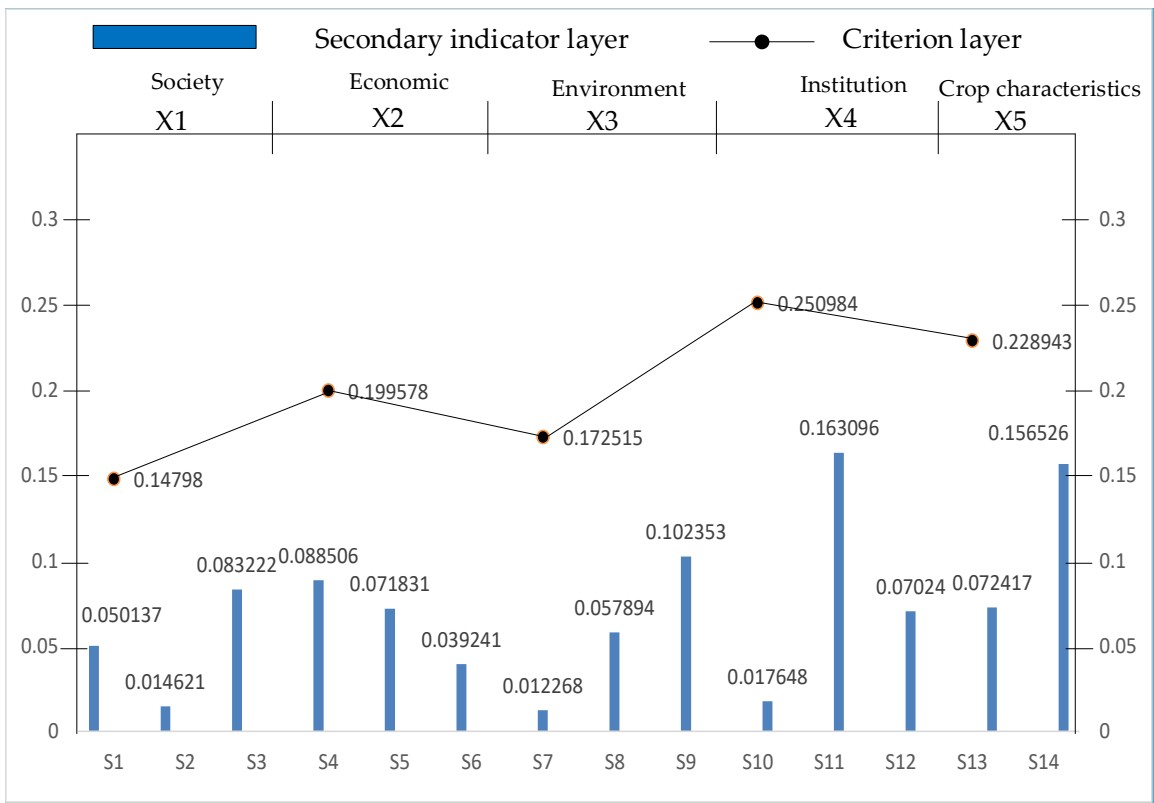

**Figure 7.** The total weight of factors of agricultural supply chain resilience to water.

As can be seen from Figure 7, among the dimension affecting the resilience of water resource management in the agricultural supply chain, institution (X4) is the main factor, with a weight of 0.250984, followed by crop characteristics (X5), accounting for 0.228943, followed by economic (X2), accounting for 0.199578, and finally environment (X3) and society (X1), respectively, accounting for 0.172515 and 0.14798. In each dimension, the main influential index weights are established water audit control system (S11) 0.1631, selection of crops (S14) 0.15653, investment in water-saving technologies (S4) 0.08851, wastewater recovery and use (S9) 0.10235, and identification of water-use risks (S3) 0.08322. Among all the indicators, establish water audit control system (S11), selection of crops (S14), and wastewater recovery and use (S9) account for a large proportion, all exceeding 0.1. Among them, establish water audit control system (S11) accounts for the largest proportion, 0.1631, combined with the hierarchical structure model in Figure 5. Combined with the total comparative weight analysis of each index in Figure 7 and Table 6, it can be concluded that there are many factors affecting the resilience of water resources management in the agricultural supply chain, and the relationship between them is very complex. The specific analysis of each dimension is shown in Figure 8.

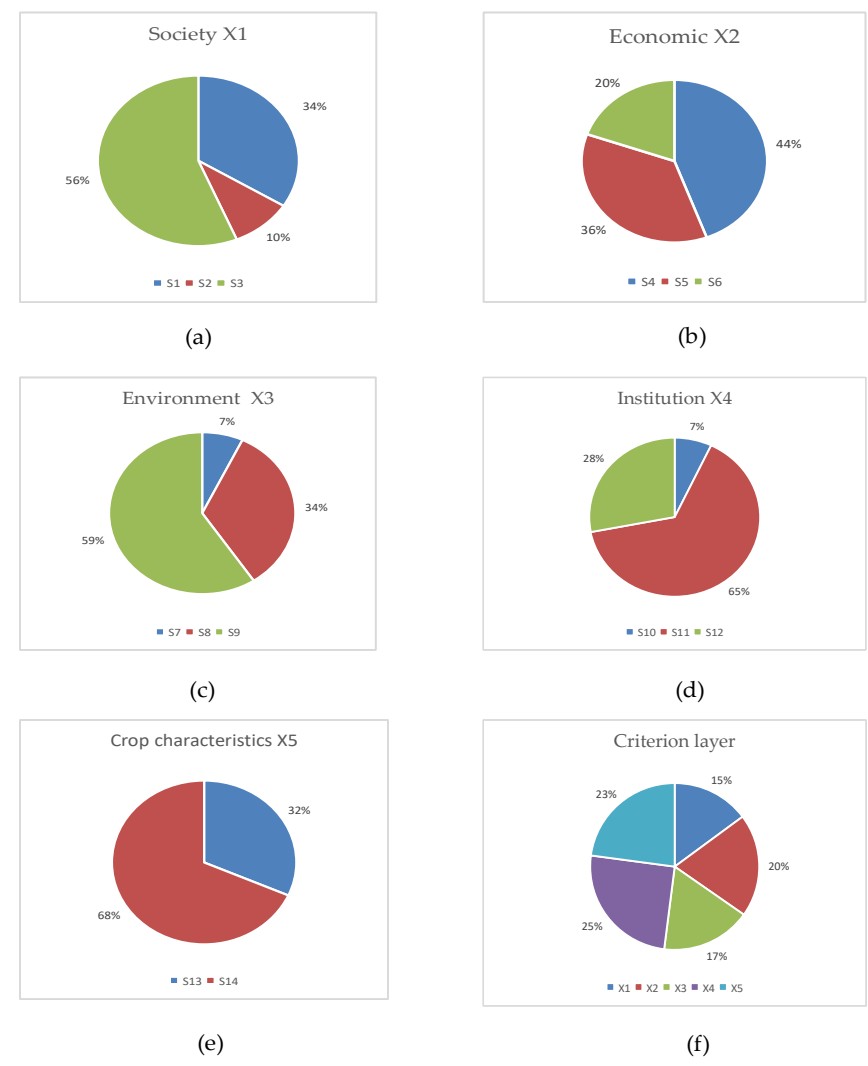

**Figure 8.** The weight ratio of influencing factors of each dimension. (**a**) Social dimension; (**b**) economy dimension; (**c**) environment dimension; (**d**) institution dimension; (**e**) crop characteristics dimension; (**f**) criterion layer dimension.

Figure 8 shows the weight ratio of index influencing factors in each dimension, including a comparison of each dimension. Combined with the hierarchical relationship of each index factor in the ISM model, it can be seen that the factors directly affect the elasticity of water resources management in the agricultural supply chain. This suggests that in the face of drought, we should consider a wide range of issues to help develop water management capacity for agricultural products. Among the factors affecting the resilience management of water resources in the agricultural supply chain, institution (X4) accounts for the highest proportion, which is 25%. In the institution (X4) dimension, the key is to consider how to establish an appropriate establish water audit control system (S11), which accounts for the largest proportion, up to 65%. The establishment of a water audit control system (S11) is also the one with the largest weight among all indicators. In addition, the crop characteristics (X5) are also an important dimension, which accounts for 23%, with the selection of crops (S14) accounting for as much as 68%. Selection of crops (S14), including the selection of crops that use less water and are more drought-tolerant, is also important in establishing robust crop characteristics, which directly affect water consumption. The economic (X2) dimension takes up 20%, with the investment in water-saving technologies (S4) taking up the highest proportion (44%), which is also a direct factor af-

fecting the resilience of agricultural water resources management. The investment in water-saving technology has a positive effect on local water resources management and enhances its ability to resist drought. The environment (X3) dimension accounts for 17%, with 59% in wastewater recovery and use (S9). In indirect factors, wastewater recovery and use (S9) reflect the importance of local water resources management. In general, wastewater reuse has a positive impact on local water resource management, thereby improving local drought resistance. Identification of water-use risk is helpful to cultivate water management awareness and enable agricultural supply chains to have standardized self-organization and self-adaptation abilities. Society (X1) accounts for 15%, with the identification of water-use risks (S3) accounting for 56%. Identification of water-use risks (S3) is helpful in cultivating water management awareness and enabling agricultural supply chains to have standardized self-organization and self-adaptation abilities. Among these basic factors, the establishment of a water audit control system is often neglected. Studies have shown that water audits can better quantify water management and lead to more rational distribution of water resources among different industries. Therefore, water resources management should be guided by these water management systems. Based on the rational allocation of water resources to agriculture, the water resources in different stages of the agricultural supply chain should be managed pertinently so as to improve the overall elasticity of water resources management in the agricultural supply chain.

In addition, investments in water-saving technologies include not only drip irrigation technologies but also rainwater harvesting systems, especially in arid areas of Northwest China. The establishment of suitable water-saving systems in the region can not only directly reduce irrigation wastewater but also collect rainfall in the region, increase the capacity of water resources at the source, and minimize the region's dependence on water resources. Therefore, the establishment of a resilient agricultural supply chain requires the government and suppliers to take critical and targeted management measures, based on the basis of a comprehensive understanding of the composition of individual factors affecting water management in the agricultural supply chain and their combined impact on water, a key resource.

## 4. Conclusions

This study uses a quantitative approach to investigate the factors influencing the-resilience of water resource management in the agricultural supply chain and proposes an integrated model. In order to consider the interaction network among various factors of the water resource management, a structural model was used to establish the hierarchical structure of water management resilience in the agricultural supply chain. The main influencing factors of water resources management in the agricultural supply chain were determined, including crop selection, water audit control system, wastewater recycling, and investment in water-saving technology. This model can effectively reflect the focus of improving the resilience of water resources management in agricultural supply chains. This water resource management resilience assessment method can be applied to the group decision-making method in agricultural supply chain management and can also be used to determine the interdependence among the key factors affecting the resilience of agricultural water resources. Some main conclusions can be drawn.

1. The model combined with the analytical network process method and interpretative structural model can be used to analyze the relationship between the factors affecting the resilience of water resources management in the agricultural supply chain. The interpretative structural model (ISM) was then used to build a three-level evaluation network. Surface direct factors include investment in water-saving technologies, reduction in waste, careful use of chemical agents, improved water retention in soils, public sector water management policies, crop selection; Indirect factors include the recycling of wastewater, the improvement of traditional crops, the setting of reason-

able water prices, and the identification of water risk; The basic factors include stakeholder participation, awareness of water conservation, the establishment of a water audit control system, and integration of water management into corporate strategy, reflecting the root and nature of the problems affecting the resilience of water management in the agricultural supply chain.

2.  A network analysis method was used to calculate the weight of each factor. The establishment of water audit control system in the system is the main factor, followed by crop characteristics, accounting for 0.23. Crops, established water audit control systems, the choice of wastewater recycle use, and impact on water-saving technology investment are the main factors of agricultural water management resilience in the supply chain.

3.  This research used the ISM method and analytic network process to comprehensively and systematically consider the agricultural water management of supply chain resilience. The mutual influence between the evaluation index and the importance of every index was used to determine the causal relationship between influencing factors. This provides a more scientific analytical framework for the development of agricultural supply chain water resources management ability in Northwest China. Furthermore, this also provides beneficial guidance for practitioners involved in agricultural supply chain management and the effective allocation of water resources.

Each index was quantified, and the weight of each index in each dimension was calculated by using the ANP method, and the resilience of water resources management in the agricultural supply chain was evaluated. According to the results, the five dimensions of agricultural supply chain management of water resources have varying significance on resilience. This suggests that, when making the appropriate interventions, measures need to be considered and weighted by the proportion of different dimensions to effectively improve the resilience of water resources management in the agricultural supply chain. According to earlier studies on arid areas in China, most of the areas have similar management systems and environmental characteristics. Therefore, the procedures identified in this study can be incorporated into a new approach to promote resilience assessment of water management in the agricultural supply chain through multiple indicators. Further, these findings and are generally applicable to other regions with similar levels of economic development, climatic characteristics, and management systems. However, the main factors affecting the resilience of agricultural supply chain water resources management will change with the development of the economy, policies and institutions, climate and environment, and other objective factors influencing the development of water-saving technology. Therefore, in future work, we will focus on the application of this evaluation method. It is necessary to apply the ISM-ANP model to establish a resilience framework for water management in the agricultural supply chain in order to help governments, farmers, and agricultural supply chain companies develop preventive, early warning, and mitigation measures using predictive analysis techniques.

**Author Contributions:** All authors were involved in the production and writing of the manuscript. Supervision, D.P.; Project administration, W.X. and Z.Z.; Data collection, S.X. and Y.Z. All authors have read and agreed to the published version of the manuscript.

**Funding:** The research was funded by the National Natural Science Foundation of China (grant number, 71503194); the Youth Foundation of Education Department of Hubei Province (grant number, B2020312); and the Centre for Service Science and Engineering Foundation of WUST (grant number, CSSE2017GA02).

**Institutional Review Board Statement:** Not applicable.

**Informed Consent Statement:** Informed consent was obtained from all subjects involved in the study.

**Data Availability Statement:** Some or all data, models, or code that support the findings of this study are available from the corresponding author upon reasonable request.

**Conflicts of Interest:** The authors declare that they have no conflicts of interest.

## Appendix A. Questionnaire for Water Resilience Management in Agricultural Supply Chain Evaluation Index

Dear Experts,

We are conducting a multi-dimensional evaluation study on the resilience of water management in the agricultural supply chain in Shaanxi and Ningxia in Northwest China. We sincerely invite you to be a consultant on the subject of "Water Resilience Management in Agricultural Supply Chain Evaluation Index". Please provide valuable opinions and suggestions for the selection of the index system during your busy schedule. The research group has selected the preliminary indicators through literature induction. The main content of this expert consultation is to evaluate and score the primary indicators in terms of importance.

The purpose of this research is to evaluate the resilience of water management in the agricultural supply chain, Shaanxi and Ningxia in Northwest China for empirical research and analyze the selected research areas based on the evaluation results and provide reasonable policy recommendations.

If you reply within 10 days, we will be very grateful!

All the members of the research group. 21 January 2021

Directions for the Application Form:

1. The following is the indicator system initially determined in our research. Please rate the importance of the indicators. Each item is divided into five levels according to the importance. They are 5 = most important, 4 = very important, 3 = medium important, 2 = not important, 1 = least important. Please rate the relative importance of the indicators and tick the corresponding □.
2. If you think this indicator is not needed, you can mark "delete" in the edit column.
3. If you think the description of the indicator is incorrect, please modify it in the content modification column.
4. For additional indicators, please fill in the blanks.

| Primary Indicators | Primary Indicators | Content Modification | Significance 1 2 3 4 5 |
|---|---|---|---|
| Society | Water-saving awareness (S1) | | □ □ □ □ □ |
| | Reduce waste (S2) | | □ □ □ □ □ |
| | Identification of water-use risks (S3) | | □ □ □ □ □ |
| Economic | Investment in water-saving technologies (S4) | | □ □ □ □ □ |
| | Stakeholder involvement (S5) | | □ □ □ □ □ |
| | Reasonable water price (S6) | | □ □ □ □ □ |
| Environment | Use chemicals with caution (S7) | | □ □ □ □ □ |
| | Improve water retention in the soil (S8) | | □ □ □ □ □ |
| | Wastewater recovery and use (S9) | | □ □ □ □ □ |
| Institution | Public sector water management policy (S10) | | □ □ □ □ □ |
| | Establish water audit control system (S11) | | □ □ □ □ □ |
| | Integrate water management into business strategy (S12) | | □ □ □ □ □ |
| Crop factors | Improving traditional crops (S13) | | □ □ □ □ □ |
| | Selection of crops (S14) | | □ □ □ □ □ |

**Table A1.** Unweighted supermatrix for compatibility before the convergence of dimensions.

| | S13 | S14 | S10 | S11 | S12 | S7 | S8 | S9 | S1 | S2 | S3 | S4 | S5 | S6 |
|---|---|---|---|---|---|---|---|---|---|---|---|---|---|---|
| S13 | 0.3846 15 | 0.2727 27 | 0.3571 43 | 0.2222 22 | 0.2108 11 | 0.3160 08 | 0.4065 04 | 0.2843 14 | 0.3333 33 | 0.3778 41 | 0.3451 93 | 0.4572 59 | 0.3372 78 | 0.42 |
| S14 | 0.6153 85 | 0.7272 73 | 0.6428 57 | 0.7777 78 | 0.7891 89 | 0.6839 92 | 0.5934 96 | 0.7156 86 | 0.6666 67 | 0.6221 59 | 0.6548 07 | 0.5427 41 | 0.6627 22 | 0.58 |

| | | | | | | | | | | | | | |
|---|---|---|---|---|---|---|---|---|---|---|---|---|---|
| S10 | 0.082342 | 0.086822 | 0.069548 | 0.078661 | 0.072399 | 0.086105 | 0.064714 | 0.073522 | 0.007167 | 0.066265 | 0.059574 | 0.053282 | 0.065281 | 0.068516 |
| S11 | 0.602629 | 0.628689 | 0.708281 | 0.669428 | 0.668913 | 0.669514 | 0.676489 | 0.673991 | 0.68863 | 0.663114 | 0.584193 | 0.689882 | 0.65866 | 0.625286 |
| S12 | 0.315029 | 0.28449 | 0.222172 | 0.251911 | 0.258688 | 0.244381 | 0.258797 | 0.252487 | 0.304203 | 0.270621 | 0.356233 | 0.256836 | 0.276058 | 0.306198 |
| S7 | 0.109309 | 0.100169 | 0.080224 | 0.062425 | 0.070048 | 0.070375 | 0.066003 | 0.069762 | 0.072579 | 0.069301 | 0.052242 | 0.062187 | 0.062259 | 0.053988 |
| S8 | 0.42367 | 0.258953 | 0.272729 | 0.351839 | 0.318469 | 0.331677 | 0.310853 | 0.320888 | 0.325276 | 0.316075 | 0.392228 | 0.352575 | 0.348573 | 0.355456 |
| S9 | 0.467021 | 0.640878 | 0.647486 | 0.585736 | 0.611483 | 0.597948 | 0.623143 | 0.60935 | 0.602145 | 0.614625 | 0.55553 | 0.585238 | 0.589168 | 0.590556 |
| S1 | 0.357559 | 0.372453 | 0.356134 | 0.371945 | 0.372139 | 0.25432 | 0.270838 | 0.270022 | 0.311519 | 0.332939 | 0.324906 | 0.378604 | 0.308321 | 0.394317 |
| S2 | 0.127981 | 0.101939 | 0.12674 | 0.104272 | 0.104629 | 0.128535 | 0.072723 | 0.127914 | 0.11228 | 0.136014 | 0.070853 | 0.066777 | 0.106302 | 0.059301 |
| S3 | 0.51446 | 0.525608 | 0.517126 | 0.523783 | 0.523232 | 0.61713 | 0.656439 | 0.602064 | 0.576201 | 0.531047 | 0.604241 | 0.554619 | 0.585376 | 0.546382 |
| S4 | 0.527541 | 0.408604 | 0.414457 | 0.534904 | 0.410892 | 0.402606 | 0.462705 | 0.438241 | 0.380035 | 0.412255 | 0.408205 | 0.409657 | 0.410711 | 0.418557 |
| S5 | 0.33233 | 0.388483 | 0.38701 | 0.270758 | 0.380885 | 0.397398 | 0.323798 | 0.411897 | 0.370504 | 0.363615 | 0.383373 | 0.400202 | 0.368686 | 0.353359 |
| S6 | 0.140129 | 0.202913 | 0.198533 | 0.194338 | 0.208223 | 0.199997 | 0.213497 | 0.149889 | 0.249461 | 0.224613 | 0.208422 | 0.190124 | 0.220603 | 0.228084 |

**Table A2.** Weighted supermatrix for compatibility before the convergence of dimensions.

| | S13 | S14 | S10 | S11 | S12 | S7 | S8 | S9 | S1 | S2 | S3 | S4 | S5 | S6 |
|---|---|---|---|---|---|---|---|---|---|---|---|---|---|---|
| S13 | 0.10763 | 0.076298 | 0.075152 | 0.046761 | 0.04436 | 0.066319 | 0.085311 | 0.059667 | 0.076188 | 0.086361 | 0.078899 | 0.096347 | 0.071067 | 0.088497 |
| S14 | 0.17216 | 0.203462 | 0.135274 | 0.163665 | 0.166066 | 0.143545 | 0.124553 | 0.150197 | 0.152377 | 0.142204 | 0.149666 | 0.114359 | 0.13964 | 0.12221 |
| S10 | 0.02633 | 0.027762 | 0.014998 | 0.016963 | 0.015612 | 0.019382 | 0.014567 | 0.016565 | 0.001728 | 0.015979 | 0.014366 | 0.013118 | 0.016073 | 0.016869 |
| S11 | 0.192697 | 0.201003 | 0.152736 | 0.144358 | 0.144247 | 0.150708 | 0.152278 | 0.151716 | 0.166055 | 0.159902 | 0.140871 | 0.169853 | 0.162166 | 0.153949 |
| S12 | 0.100734 | 0.090969 | 0.047901 | 0.054323 | 0.055784 | 0.05501 | 0.058255 | 0.056835 | 0.073355 | 0.065257 | 0.085901 | 0.063234 | 0.067967 | 0.075388 |
| S7 | 0.014592 | 0.013372 | 0.015348 | 0.011942 | 0.013401 | 0.013362 | 0.012532 | 0.013246 | 0.012686 | 0.012113 | 0.009131 | 0.011005 | 0.011017 | 0.009554 |
| S8 | 0.056557 | 0.034568 | 0.052092 | 0.06731 | 0.060926 | 0.062975 | 0.059021 | 0.060926 | 0.056853 | 0.055245 | 0.068556 | 0.062392 | 0.061684 | 0.062902 |
| S9 | 0.062344 | 0.085553 | 0.12387 | 0.112057 | 0.116983 | 0.113531 | 0.118315 | 0.115696 | 0.105246 | 0.107427 | 0.097099 | 0.103564 | 0.104259 | 0.104505 |
| S1 | 0.023866 | 0.02486 | 0.065035 | 0.067922 | 0.067958 | 0.044526 | 0.047418 | 0.047275 | 0.048166 | 0.051478 | 0.050236 | 0.064085 | 0.052188 | 0.066745 |
| S2 | 0.008542 | 0.006804 | 0.023144 | 0.019041 | 0.019107 | 0.022506 | 0.012732 | 0.022395 | 0.017361 | 0.02103 | 0.010955 | 0.011303 | 0.017993 | 0.010038 |
| S3 | 0.034338 | 0.035082 | 0.094434 | 0.09565 | 0.095549 | 0.108046 | 0.114928 | 0.105408 | 0.089091 | 0.082109 | 0.093426 | 0.093878 | 0.099084 | 0.092484 |
| S4 | 0.105634 | 0.081819 | 0.082894 | 0.106984 | 0.082181 | 0.080557 | 0.092582 | 0.087687 | 0.076346 | 0.082819 | 0.082006 | 0.080646 | 0.080853 | 0.082398 |
| S5 | 0.066546 | 0.07779 | 0.077404 | 0.054153 | 0.076179 | 0.079515 | 0.064789 | 0.082411 | 0.074432 | 0.073048 | 0.077017 | 0.078785 | 0.07258 | 0.069563 |
| S6 | 0.028059 | 0.040631 | 0.039708 | 0.038869 | 0.041646 | 0.040017 | 0.042719 | 0.029991 | 0.050115 | 0.045026 | 0.041871 | 0.037431 | 0.043428 | 0.044901 |

**Table A3.** Supermatrix for compatibility after convergence of dimensions.

| | S13 | S14 | S10 | S11 | S12 | S7 | S8 | S9 | S1 | S2 | S3 | S4 | S5 | S6 |
|---|---|---|---|---|---|---|---|---|---|---|---|---|---|---|
| S13 | 0.072417 | 0.072417 | 0.072417 | 0.072417 | 0.072417 | 0.072417 | 0.072417 | 0.072417 | 0.072417 | 0.072417 | 0.072417 | 0.072417 | 0.072417 | 0.072417 |
| S14 | 0.156526 | 0.156526 | 0.156526 | 0.156526 | 0.156526 | 0.156526 | 0.156526 | 0.156526 | 0.156526 | 0.156526 | 0.156526 | 0.156526 | 0.156526 | 0.156526 |
| S10 | 0.017648 | 0.017648 | 0.017648 | 0.017648 | 0.017648 | 0.017648 | 0.017648 | 0.017648 | 0.017648 | 0.017648 | 0.017648 | 0.017648 | 0.017648 | 0.017648 |
| S11 | 0.163096 | 0.163096 | 0.163096 | 0.163096 | 0.163096 | 0.163096 | 0.163096 | 0.163096 | 0.163096 | 0.163096 | 0.163096 | 0.163096 | 0.163096 | 0.163096 |
| S12 | 0.07024 | 0.07024 | 0.07024 | 0.07024 | 0.07024 | 0.07024 | 0.07024 | 0.07024 | 0.07024 | 0.07024 | 0.07024 | 0.07024 | 0.07024 | 0.07024 |
| S7 | 0.012268 | 0.012268 | 0.012268 | 0.012268 | 0.012268 | 0.012268 | 0.012268 | 0.012268 | 0.012268 | 0.012268 | 0.012268 | 0.012268 | 0.012268 | 0.012268 |
| S8 | 0.057894 | 0.057894 | 0.057894 | 0.057894 | 0.057894 | 0.057894 | 0.057894 | 0.057894 | 0.057894 | 0.057894 | 0.057894 | 0.057894 | 0.057894 | 0.057894 |
| S9 | 0.102353 | 0.102353 | 0.102353 | 0.102353 | 0.102353 | 0.102353 | 0.102353 | 0.102353 | 0.102353 | 0.102353 | 0.102353 | 0.102353 | 0.102353 | 0.102353 |
| S1 | 0.050137 | 0.050137 | 0.050137 | 0.050137 | 0.050137 | 0.050137 | 0.050137 | 0.050137 | 0.050137 | 0.050137 | 0.050137 | 0.050137 | 0.050137 | 0.050137 |
| S2 | 0.014621 | 0.014621 | 0.014621 | 0.014621 | 0.014621 | 0.014621 | 0.014621 | 0.014621 | 0.014621 | 0.014621 | 0.014621 | 0.014621 | 0.014621 | 0.014621 |
| S3 | 0.083222 | 0.083222 | 0.083222 | 0.083222 | 0.083222 | 0.083222 | 0.083222 | 0.083222 | 0.083222 | 0.083222 | 0.083222 | 0.083222 | 0.083222 | 0.083222 |
| S4 | 0.088506 | 0.088506 | 0.088506 | 0.088506 | 0.088506 | 0.088506 | 0.088506 | 0.088506 | 0.088506 | 0.088506 | 0.088506 | 0.088506 | 0.088506 | 0.088506 |
| S5 | 0.071831 | 0.071831 | 0.071831 | 0.071831 | 0.071831 | 0.071831 | 0.071831 | 0.071831 | 0.071831 | 0.071831 | 0.071831 | 0.071831 | 0.071831 | 0.071831 |
| S6 | 0.039241 | 0.039241 | 0.039241 | 0.039241 | 0.039241 | 0.039241 | 0.039241 | 0.039241 | 0.039241 | 0.039241 | 0.039241 | 0.039241 | 0.039241 | 0.039241 |

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
