# Peer review of "Enhancing the Resilience of the Management of Water Resources in the Agricultural Supply Chain"

_water, doi:10.3390/w13121619_

Round 1

Reviewer 1 Report

The subject is current and very important. This paper proposes the use of identify the factors affecting the resilience of water management in the agricultural supply chain and to help manage the risks related to water resources utilization. The following factors were selected for the analyzes: society, economy, environment, institution and crop characteristics.  A quantitative assessment of the resilience of the management of water resources has not been introduced into the agricultural supply chain, and there is also a lack of a comprehensive framework for the management of water resources in the entire agricultural supply chain. Using a combination of an interpretative structural model (ISM) and an analytical network process (ANP), a hierarchical structure model was developed, composed of direct factors, indirect factors and basic factors.
Unfortunately, the work has a number of weaknesses that must be corrected before publication.
1. The layout of the work is incorrect. Point 1.1. is unnecessary. The clearly indicated chapter Results is missing. Their analysis is also only partial. The Discussion chapter is completely missing, where one should refer to the results of other studies (other researchers).
2. Regarding the method, it is shown that “it works”. But the main concern is that it is only useful, as presented, in a single and particular case of data. This is not enough to justify the proposal. Moreover, some methodological aspects are not clear.
3. Figures 4, 7 and 8 are unclear, illegible, poorly described.
4. Referring to 41 works meets a certain minimum. But, as already mentioned, one can, for example, refer to more current works in the Discussion.

The reviewer conclusion is that, the overall presentation should be improved, and refocused. Lastly, the link with sustainability is marginal and almost implicit. It should be improved and send the article should be re-reviewed.

Reviewer 2 Report

Thank you for the opportunity to review this nicely presented piece of interesting and timely research.

The manuscript is concerned with "Enhancing the resilience of the management of water resources in the agricultural supply chain ” which is very interesting and actual. It could be with the aim and scope of the journal. The paper is technical sound and interdisciplinary in this present form, it contains only minor weaknesses which, in my opinion, is better to overcome before publication in Water.

  • Abstract & introduction: are focused on the main aim of the paper and what is the new contribution of authors to the state of art. Improving the resilience of water resource management is a key measure to reduce the risks in the agricultural supply chain. This study aims to identify the factors affecting the resilience of water management in the agricultural supply chain and to help manage the risks related to water resources utilization.
  • Materials & methods: Methods seems adequate and results are well presented. The authors combined different research methods (questionaries, ISM, AHP, ANP) for the area of agricultural industry in Northwest of China. But the suitability and technical standards of the methods are described with sufficient details of the processes so that another researcher is able to reproduce the methods for different similar areas over the world.
  • Results & discussion: The data are well controlled and robust and authors provided relevant and current references. Conclusions are strictly based on actual facts and figures. In my opinion they carried out sufficient and appropriate statistical analyses.
  • Conclusion: Authors provided adequate proof for their claims. Also they wrote about positives and negatives of their research and how it can be applicable in the practise. Their findings provide useful guidance for practitioners in the management of water resources in agricultural supply chains. The results show that the selection of agricultural products, the establishment of a water audit control institution, the recycling of wastewater and the investment in water-saving technologies are the main factors affecting the resilience of water resources management in the agricultural supply chain. These results contribute to the sustainable management and strategic deployment of water by agricultural supply chain stakeholders.

For the future study could be more informative also in the economical point of view.    

  • All the references cited are relevant and adequate.

Some editorial errors are present in the manuscript:                   

  • 1st chapter has only one subchapter 1.1 (line 194)  
  • Figures 1 and 3 has the titles at the other page
  • Table 3 and Figure 4 – Please unite the writing of the first letter (Water.saving,..).
  • Line 372 why Among with big letter in the middle of the sentence?
  • More times authors don´t used space between Figure and number (Figure7, ...)

It seems relevant to include the questionnaires in the Supplementary Material.

Also I cannot see more information about used Super decision software.

Reviewer 3 Report

REVIEW COMMENTS

Enhancing the resilience of the management of water resources in the agricultural supply chain

Wenping Xu, Zhi Zhong, David. Proverbs and Shu Xiong, Yuan Zhang

The manuscript of Xu et al. presents very interesting paper about modelling for guidance to practitioners involved in the management of water resources in agricul- tural supply chains for the sustainable management and strategic deployment of water by agricultural supply chain stakeholders.

I think that this paper helps researchers who want to study the management of water resources in the agricultural supply

Major comments

Dear authors, I found this article interesting and indeed highlighting a knowledge gap of importance. However, I have several remarks before the paper can published as is in Water Journal

1) I think the introduction part is too long and too general.

Minor comments

1) Figure 1, 4, 8 it is not easy to read.

Round 2

Reviewer 1 Report

Thank you very much for the new manuscript sent, which took into account, among others, my remarks and comments. The authors contributed new information to the text. Currently, the work is much better and clearer, and its subject matter is interesting and important. Taking into account its present form, I can recommend the paper for publication.

Author Response

Thank you very much for your comments on this paper. Your comments have brought us a lot of inspiration. Under your guidance, the content of the paper has been enriched, the quality has been improved, and the practical value has been highlighted. Once again, we express our sincere thanks to you!